# Multicomponent Oleogels Prepared with High- and Low-Molecular-Weight Oleogelators: Ethylcellulose and Waxes

**DOI:** 10.3390/foods12163093

**Published:** 2023-08-17

**Authors:** Ziyu Wang, Jayani Chandrapala, Tuyen Truong, Asgar Farahnaky

**Affiliations:** Biosciences and Food Technology, School of Science, RMIT University, Melbourne, VIC 3082, Australia; s3742701@student.rmit.edu.au (Z.W.); jayani.chandrapala@rmit.edu.au (J.C.); tuyen.truong@rmit.edu.au (T.T.)

**Keywords:** oil structuring, multicomponent oleogel, ethylcellulose, beeswax, carnauba wax

## Abstract

The combined interactions between ethylcellulose (EC) and natural waxes to structure edible oil are underexplored. To reduce the high EC concentration required to form a functional oleogel, novel oleogels were prepared using a 50% critical concentration of EC (i.e., 4%) with 1–4% beeswax (BW) and carnauba wax (CRW). One percent wax was sufficient for EC to form self-sustaining oleogel. Rheological analysis demonstrated that 4%EC + 4%BW/CRW had comparable oleogel properties to 8%EC. The yield stress and flow point of wax oleogels were enhanced upon EC addition. EC did not influence the thermal behaviour of the wax component of the oleogel, but the crystallinity and plasticity of the combined oleogel increased. The crystal shape of BW oleogel changed upon EC addition from a needle-like to spherulitic shape. Confocal laser scanning microscopy highlighted the uniform distribution of EC polymeric network and wax crystals. EC/wax mixtures have promising oil-structuring abilities that have the potential to use as solid fat substitutes.

## 1. Introduction

With increasing awareness of health concerns related to saturated and *trans* fats (e.g., risk of heart disease, diabetes, etc.), the World Health Organization has stated that the intake of saturated and *trans* fats should be less than 10% and 1% of total energy intake, respectively. Therefore, fat consumption needs to shift toward unsaturated fat [1]. However, the fat-crystal network resulting from solid fats is responsible for many functional and organoleptic properties of processed foods. Therefore, replacing solid fats with liquid vegetable oil that is high in unsaturated fatty acids may result in many complications relating to processing and product quality.

One of the most promising strategies to replace fat crystal networks is to create a healthier “alternative fat network” through oleogelation [2]. Generally, this physical modification method involves a simple procedure of heating and cooling oleogelators in liquid oil [3]. The gelator becomes self-assembled through cooling below the gelation transition temperature and transforming into a thermoreversible three-dimensional network that entraps the surrounding oil; an illustration of the main oleogel-preparation methods can be found in the review paper published by our research group [4]. Low-molecular-weight gelators (LMWGs) (<3000 Da), including waxes, monoglycerides, fatty acids, alcohols, Span 60, and combinations of phytosterols and oryzanol, are widely studied due to their ease of preparation (direct dispersion), high gelling capacity, and high availability. More recently, researchers demonstrated that proteins and polysaccharides (polymeric gelators) can successfully produce oleogels but require multiple, more complicated preparation steps (such as emulsion and the foam-templated approach) due to their hydrophilic nature [5], which may not be preferred on an industrial scale. For this reason, ethylcellulose (EC), the only known polymeric gelator that can directly dissolve in oils, is regarded as the most promising gelator.

EC is a linear polysaccharide synthesised from cellulose when some of the functional hydroxyl groups on the repeating glucose units are substituted with ethylene groups along the polymer backbone [6]. It must have a degree of substitution between 2.4 and 2.5 to be dissolved in organic solvents. When EC is heated above its glass-transition temperature, the strands of EC become flexible and rubbery and allow them to interact interact with solvents, leading to dissolution; upon cooling, the liquid oil is physically entrapped within the 3D polymer network as a result of the formation of inter- and intramolecular junction zones among EC macromolecules [7]. Previous studies on EC oleogels have covered various aspects of their properties, such as the effect of the EC’s molecular weight [8], the impact of the oil composition [7,9] the thermo-oxidative behaviour [10], the responsiveness to digestive lipolysis [11] and the stability of bioactive compounds [12]. The incorporation of EC oleogel into food matrices has also been achieved in cream cheese [13], shortening [14], sausages [15], and beef burgers [16]. However, EC oleogel also shows several technical limitations, including the high concentration required to gel oils (>6% *w/w*), its high melting temperature (>130 °C), its poor plasticity, and its difficulty in spreading. A common strategy to improve or overcome these limitations is introducing a secondary gelator.

Combinations of EC and LMWGs have demonstrated many possibilities for improving oleogels’ properties. Adding monoglycerides to the EC system enhanced the complex modulus of the combined oleogel and reduced oil loss [17]. Interactions between EC and stearyl alcohol (SO) and stearic acid (SA) reduced the brittle nature of SOSA oleogel when the EC concentration was below its critical gelling concentration, enhancing its plasticity and flow properties [18]. However, natural waxes, as one of the most promising LWMGs, have not been systemically studied in conjunction with EC. Waxes are chemically heterogenous substances composed of long-chain esters generated from fatty acids and alcohols. They are widely available and are able to structure oil at low concentrations (0.5–4% *w/w*), and many types of wax are agricultural by-products. [19] highlighted that EC can interact with wax to produce heat-resistant chocolates, but no in-depth experiments were performed since wax was not the focus of their study. A more recent study by [20] investigated the rheological properties of EC–monoglyceride–candelilla wax oleogel. The authors found that monoglyceride and candelilla wax crystals improved the consistency of oleogel. However, the sole interaction between EC and candelilla wax was not studied. Therefore, there is a need to understand the mechanism of interaction between EC and waxes, which could play a pivotal role in overcoming the functional limitations of these two individual gelators and broaden the functional properties of the new oleogel systems. Considering the ability of natural waxes to form oleogels at low concentrations, we hypothesise that the addition of waxes to EC oleogel systems could reduce the EC concentration required to form a functional oleogel, therefore reducing the high cost and consumption of EC as a chemically modified cellulose derivative. BW and CRW were selected as the model waxes for the current study due to their different chemical compositions and gelling efficiencies, their affordable price, and the wide availability of information on them in the literature. Therefore, the objective of the current study is to develop novel combined oleogels by using a low concentration of EC (50% of its critical concentration) in combination with BW or CRW. The knowledge gained from this study will provide a better understanding of oleogels with combined EC and LMWGs, specifically the interactions between EC and waxes, to develop low-fat and functional food products.

## 2. Materials and Methods

### 2.1. Materials

Rice bran oil (RBO), BW (contains mainly esters and hydrocarbons), refined CRW (extracted from palm leaves and containing aliphatic esters, w-hydroxy esters, and unsaturated alcohols), and ethylcellulose (EC; 22 cP, 5% in toluene/ethanol 80:20 at 25 °C, softening temperature 155 °C) were purchased from a local supermarket, Honey Joy (Melbourne, VIC, Australia), and Sigma-Aldrich (Castle Hill, Australia), respectively.

### 2.2. Preparation of Oleogels

Twelve oleogels were prepared using commercial RBO (Table 1), and at least three replicates were conducted for each sample. The critical concentration of oleogelators depends on a number of factors, including the oil type, oleogel type, preparation method, storage temperature, and time. Since each gelator has different gelling mechanisms and capacities, the critical concentration of each gelator at room temperature (8%, 3%, and 6% *w/w*, respectively) was selected as the control concentration instead of choosing a constant concentration for all gelators. These critical concentrations were obtained by preparing a series of oleogels using a range of concentrations of each of these oleogelators and testing their ability to create oleogels after overnight storage. A 4% *w/w* EC oleogel (half of the critical concentration of EC, denoted as 4EC) was prepared to determine the impact of the inclusion of waxes on combined EC + wax oleogels. In order to avoid the development of an unpleasant waxy mouthfeel in the combined oleogel systems, the highest wax concentrations chosen for both waxes were 4% *w/w*. Thus, the treatment group included 4% *w/w* EC in conjunction with 1, 2, 3, or 4% *w/w* BW or CRW.

For the control samples, the gelators were mixed with RBO on a hot plate for 30 min with gentle stirring (200 rpm) at 80 °C for BW, 90 °C for CRW, and 170 °C for EC. These temperatures were selected to be around 20 °C above each gelator’s melting or softening temperatures. Then, the samples were allowed to cool statically either at a refrigeration temperature (4 °C) or at room temperature (25 °C), and then they were stored at these temperatures. We investigated two different temperatures with the aim of providing insight into the properties of oleogel in applications that go through cold and room-temperature storage.

For the treatment groups, 4% *w/w* EC was first allowed to dissolve in RBO by heating and mixing at 170 °C for about 30 min. Then, the EC-RBO mixture was transferred to a water bath (pre-heated at 80 °C or 90 °C) to cool down statically. Once the temperature reached 80 °C (for EC + BW oleogel) or 90 °C (for EC + CRW oleogel), BW or CRW were added at desired concentrations. The mixture was stirred for another 10 min; then, the samples were cooled statically to either 4 or 25 °C and stored at these temperatures. All samples were stored for 24 h prior to further analysis. The cooling rate (°C min^−1^) of samples was monitored and recorded using a digital data logger and was calculated as the rate at each data point, and the average value was computed.

### 2.3. Characterisation of Oleogels

#### 2.3.1. Visual Appearance and Morphological Analysis

After sample preparation, approximately 5 mL of each oleogel sample was poured into centrifuge tubes; then, the samples were cooled down and stored at either 4 or 25 °C. The tubes were inverted after 24 h and examined visually in terms of gel formation and appearance. The crystal morphology of oleogels was analysed using a polarised light microscope (PLM) with a digital camera (ECLIPSE Ci POL—Nikon, Japan). After 24 h of storage, a tiny amount of oleogel was placed on a glass slide, covered with a coverslip and observed at room temperature. The images of wax crystalline structures were observed at a magnification of 20×, and the images were analysed using the Image J software, version 2 (Fiji, Washington, DC, USA) [21]. Selected oleogels were also prepared for confocal laser scanning microscopy (CLSM) analysis; the oleogel samples were initially prepared in the same way as those described in Section 2.2, and then Nile red (0.1%) was added during the last 10 min of sample stirring and stored in the dark at 25 °C for 24 h. Samples were imaged using a Nikon Inverted Microscope Eclipse Ti-E (Tokyo, Japan) fitted with A1R laser at a magnification of 20× at excitation and emission wavelength of 561 and 570–620 nm, respectively.

#### 2.3.2. Dynamic Rheology Measurements

All rheological measurement was carried out using a TA Instruments Discovery Hybrid Rheometer equipped with a Peltier system (New Castle, DE, USA). Parallel-plate geometry (40 mm diameter) with a gap of 1000 μm was used. The isothermal measurements were performed at either 4 or 25 °C, depending on the storage condition of oleogel.

##### Amplitude Sweep

Amplitude sweeps were performed at 0.01–100% strain at a constant frequency of 1Hz to determine the linear viscoelastic region (LVR), storage (G′), and loss (G″) modulus of each oleogels. Yield stress and flow point were also recorded.

##### Frequency Sweep

Frequency sweeps were performed with a strain value within the LVR and a frequency range of 0.01 to 10 Hz. G′ and G″ were recorded to evaluate the frequency-dependent behaviour of the oleogels.

##### Thixotropy Behaviour

Oleogel samples were subjected to consecutive steps of low (0.1 s^−1^), high (10 s^−1^), and low (0.1 s^−1^) shear rates for 3, 3, and 15 min, respectively. The G′ obtained after the first shear is the initial G′, whereas the G′ at the end of the treatment was the recovery G′. The recovery (%) is calculated by dividing G′ at the 3rd interval by G′ at the end of the 1st interval.

##### Temperature Rampng

Temperature ramping was performed with a strain value within the LVR at a constant frequency (1 Hz). The oleogel samples were heated from either 4 or 25 °C to 130 °C at a rate of 2 °C min^−1^. The G′-G″ crossing-over temperatures of oleogel samples were recorded.

#### 2.3.3. Texture Measurement

The texture of oleogels was analysed using a TA.XT.plus texture analyser (Stable Micro Systems, Surrey, England). After the sample preparation (Section 2.2), a small amount of each hot sample was poured into a Plexiglass mould (20 mm diameter, 10 mm height). The samples were left to set at 4 or 25 °C for at least 24 h. The hardness was measured using a 35 mm diameter cylindrical probe to a depth of 5 mm at a speed of 1 mm s^−1^. Textural parameters (hardness, adhesiveness, cohesiveness, and gumminess) were measured from the force–time/distance graph.

#### 2.3.4. Oil-Binding Capacity (OBC)

After sample preparation, approximately 1 g of oleogel was weighed in a 1 mL centrifuge tube. It was then stored at 4 and 25 °C for 24 h. The sample was centrifuged at 2000 or 10,000 rpm at 4 or 25 °C for 15 min. The tube was then inverted for 10 min on a qualitative filter paper (Whatman #1), and oil loss was determined via gravimetry.

The oil loss (OL) was calculated according to [22]:OL = [(m_1_ − m) − (m_2_ − m)]/(m_1_ − m) × 100%
OBC% = 100 − OL
where m is the mass of the centrifuge tube, m_1_ is the mass of the centrifuge tube with initial sample, and m_2_ is the mass of the centrifuge tube with sample mass after centrifugation [22].

#### 2.3.5. Differential Scanning Calorimetry (DSC)

The thermal profiles of oleogels were examined using a modulated DSC using a TA Instrument, Q2000 (New Castle, DE, USA). Oleogel samples (6–8 mg) were sealed hermetically in an aluminium Tzero pan. BW, CRW, and their corresponding oleogels were heated from 0 to 120 °C; EC powder, RBO, and the combined oleogels were heated from 0 to 250 °C at a rate of 10 °C min^−1^. The samples were held isothermally for 5 min to eliminate any thermal history. In the subsequent step, the samples were cooled down from 250 °C or 120 °C to 0 °C at 10 °C min^−1^. The second heating and cooling cycles were performed at the same scan rates. Onset (T_onset_), peak (T_peak_), endset temperature (T_endset,_ °C), and enthalpy (ΔH, J/g) during the heating and cooling were obtained using the software TA Instruments Universal Analysis Version 4.5 (New Castle, DE, USA), provided by the instrument supplier. The analysis was performed in triplicate.

#### 2.3.6. X-ray Diffraction (XRD)

XRD analysis was performed using a Bruker D4 Endeavor (Karlsruhe, Germany). EC powder and oleogels were gently spread onto the sample holder and tested in that form. The pellets of BW and CRW were firstly melted in a water bath and then poured into a Plexiglass mould and allowed to cool down. Raw ingredients (EC powder, BW and CRW) and oleogels were examined at intervals of 0.02° with a 2θ range from 5 to 40° at room temperature. The diffractometer was equipped with a Cu-Kα radiation source operated at 40 kV and a current of 35 mA at 1.5 Å wavelength. The diffractograms were analysed in the DIFFRAC EVA V4.2 software (Bruker AXS Materials Research Software, Karlsruhe, Germany).

#### 2.3.7. Fourier Transform Infrared (FTIR) Spectroscopy

FT-IR spectra of oleogels, EC powders, BW, and CRW were measured using the Perkin Elmer Spectrum 100 FTIR spectrometer (Norwalk, CT, USA). All oleogels and EC powder were measured directly as is. Interferograms were accumulated over a spectral range of 4000–400 cm^−1^ with a nominal resolution of 4 cm^−1^ and 60 scans.

### 2.4. Statistical Analysis

All data were analysed using the IBM SPSS Statistic 26.0 package software (IBM Corp., New York, NY, USA) by performing a one-way analysis of variance (ANOVA) and Tukey’s multiple comparison (*p* < 0.05) to determine the significant differences between the samples.

## 3. Results and Discussion

### 3.1. Visual Appearance and Morphological Properties

Figure 1a shows the visual appearance of the control and combined oleogels at room temperature. The 3BW and 6CRW oleogels exhibit an opaque appearance, whereas 8EC is a transparent solid (Figure 1a). A similar transparent appearance was also reported in EC–corn-oil oleogel [23]. As expected, 4EC did not form an oleogel, as it flows when inverted (Figure 1b). However, adding 1% *w/w* BW or CRW to the 4EC sample successfully produced a self-sustaining oleogel, and as the wax concentration increased, the oleogel became opaquer (Figure 1c,d). Preliminary work found that at least 3% *w/w* BW and 6% *w/w* CRW are required to form an oleogel at room temperature, suggesting a synergistic effect between EC + wax as an oil structurant. The presence of waxes increased the structurant presented in the 4EC oleogel, and their interactions during crystallisation led to the formation of a space-filling network and subsequent gelation. Oleogels cooled and stored at 4 °C showed similar appearances.

The PLM images of the control and combined oleogels are shown in Figure 2a. It has been suggested previously that liquid oils appear dark under PLM due to their optically isotropic characteristic [24]. Therefore, the crystals shown are associated with the oleogelators. The 3BW oleogel under both cooling and storage conditions had needle-like crystal structures. Similar observations were reported by [25,26]. The 6CRW oleogel, on the other hand, presented aggregated-like, branched dendritic crystals and loose arrangements, which have a relatively poor ability to entrap liquid oil. PLM images also show that in the absence of EC, the BW oleogel had thin needle-like crystals; however, the CRW oleogel had much thicker crystals. Moreover, the lengths of BW and CRW crystals were reduced in the presence of EC. For the 8EC oleogel (images not shown), no birefringence could be observed under the polarised light, and similar observations were also reported by [27]. Therefore, for the oleogels with combined EC and wax, PLM was used to observe the changes to wax crystals upon EC addition. The addition of EC changed the crystal shape of beeswax oleogel from a needle-like shape to a spherulitic shape. The 4EC1BW oleogel had much smaller crystals, which could be due to the low wax concentration [28]. 4EC2BW oleogels had mixed needle-like and spherulitic-shaped crystals. Furthermore, the addition of 4% EC broke the branched dendritic crystals of CRW oleogel into smaller, less branched, and more rounded crystals. As the CRW concentration increased from 1 to 4% *w/w*, the single crystals started to aggregate again, and the 4EC4CRW oleogel contained both aggregated and single crystals.

By comparing the microstructures of oleogels stored at different temperatures, it can be seen that those cooled and stored at 4 °C had relatively smaller and more densely packed and evenly distributed crystals. This may have contributed to an increased gel hardness compared to those at 25 °C. It has been suggested previously that a slower cooling rate tends to produce larger crystals due to the slow rate of onset of nucleation and crystal formation, whereas a faster cooling rate forms small crystals as a result of the early onset of crystallisation [28,29]. Nevertheless, the difference between the wax crystal sizes under different storage conditions is not distinct. In our study, the cooling rate was recorded by leaving freshly prepared oleogel samples statically at 4 °C and 25 °C and recording their temperature change, which had cooling rate values of 1.02 ± 0.02 °C min^−1^ and 0.72 ± 0.07 °C min^−1^, respectively. Compared to other studies, which had a controlled and distinct difference between cooling rates (e.g., 1 and 10 °C min^−1^ from [30], it is possible that the minor cooling rate differences in the current study did not have a pronounced influence on the crystal size. Furthermore, PLM can only be used to observe the changes in wax crystals upon EC addition. The distribution of EC particles in the combined system was not shown. Thus, the above observations can only partially explicate the properties of the combined oleogels.

Selected oleogels were also analysed using CLSM to gain a better understanding of the interactions between EC and wax in the RBO structuration (Figure 2b). Nile red was used to identify trapped oils in oleogels, and wax crystals appear dark as the dye cannot penetrate crystals. It can be seen that 3BW and 6CRW demonstrated a clear and distinct crystal morphology, with the former one appearing as needle-like crystals and the latter one appearing dendritic-like, similar to those observed in PLM. The 8EC oleogel had the presence of small, reddish-white particles (~10 µm) on top of the red background, as indicated by the white arrow, which could indicate some of the EC structure under CLSM. These reddish-white particles were also found in the combined oleogels, but these particles seemed to aggregate and form larger particles (~30–50 µm) compared to those in the 8EC oleogel. This observation might have been caused by the incorporation of wax in the system; both wax crystals and EC particles led to a space-filling network, entrapping the surrounding oil. Furthermore, the wax crystals in the combined EC + wax oleogels became much smaller than those in the wax oleogel alone. It seems that the presence of EC hinders the interactions of wax crystals, preventing them from forming larger crystals. Nevertheless, the interactions between the EC polymeric network and the wax crystals were uniform without any phase separation. Further microscopic techniques such as cryo-SEM can be used in future studies to observe the polymeric network of EC and the surface morphology of the combined oleogels, which could contribute to a more complete understanding of EC and wax interactions.

### 3.2. Dynamic Rheology Measurements

#### 3.2.1. Amplitude Sweep

Figure 3a,c show that all oleogels (stored at 25 °C) had G′ > G″, indicating solid-like behaviour (i.e., more elastic than viscous). Although the 4EC oleogel also had higher elastic than viscous modulus at strains less than 10% at both storage temperatures (4 or 25 °C), its G′ LVR value is low (~20 Pa), and the difference between its G′ and G″ values was only approximately 10 Pa. This result corroborates the visual appearance (a thick liquid) that the 4EC oleogel cannot form a strong polymeric network to entrap large amounts of oil, even though it showed the elastic component. With the addition of 1% BW to the 4EC, its G′ increased significantly from ~21 Pa to 3700 Pa. The 4EC3BW and 4EC4BW oleogels had similar G′ to 4EC1BW, suggesting that adding more BW did not contribute to a greater gel strength. Nevertheless, the 4EC2BW (G′ = ~5500 Pa) produced the strongest oleogel among all 4EC + BW samples but was softer than that of the 8EC (G′ = ~8000 Pa).

A similar trend was observed in CRW containing oleogel; that is, the inclusion of 1% *w/w* CRW to 4EC also increased the G′ (Figure 3c) to approximately 1800 Pa. The 4EC1CRW oleogel had a lower G′ compared to that of the 4EC1BW; this could be explained by the different gelling capacities of the two waxes. The lowest BW concentration (1% *w/w*) investigated can provide considerable structuring support to the RBO because 3% *w/w* BW is sufficient to produce an oleogel by itself. In contrast, 6% *w/w* is required for CRW to structure RBO. Therefore, the addition of 1% CRW to 4EC may not create an oleogel that is as strong as the 4EC + 1% BW oleogel. A similar finding was also reported by [31], who found that 1–1.5% *w/w* BW is able to structure RBO at 5 °C, whereas for CRW, 4% *w/w* is required. They explained that BW is an efficient gelator because it has a balanced solubility and insolubility in RBO; this enhanced the wax–wax and wax–oil interactions, creating a strong gel network. On the other hand, CRW had a higher solubility in RBO, preventing the formation of a strong network at low concentrations [31] In general, for all EC + CRW oleogels, the G′ increased with CRW concentration. The 6CRW oleogel had a G′ in between the 4EC2CRW and the 4EC4CRW. When oleogels were cooled and stored at 4 °C (Figure 3b,d), all samples had increased gel strength values (i.e., G′) compared to their room-temperature counterparts. Among all the combined oleogels, the 4EC4BW and the 4EC4CRW had the highest G′ with values of ~11,600 Pa and ~17,500 Pa, respectively. The 4EC4CRW oleogel had a comparable G′ to that of the 8EC (G′ = ~18,000 Pa).

Yield stress represents the end of LVR, indicating the softening point of oleogels. It was determined from the strain sweep when a 5% drop in the G′ from the average of the plateau was observed. The flow point represents the crossing over of G′ and G″, indicating that gels underwent permanent deformation. The wax-based oleogels had low yield-stress values (2.29 ± 0.69 Pa for BW and 1.82 ± 0.15 Pa for CRW) and flow points (18.55 ± 3.75 Pa for BW and 4.36 ± 0.56 Pa for CRW), indicating their shear sensitivity (Figure 3e,f). This finding is consistent with those reported by [25], where beeswax/olive-linseed-fish oil oleogel showed a solid-like but shear-sensitive gel property. EC oleogel, on the other hand, showed high resistance to applied shear at its critical concentration, with 55.59 ± 4.87 Pa for yield stress and 413.79 ± 16.31 Pa for flow point. Oleogels prepared with combined gelators had an improved yield stress (with a range from 19.53 Pa to 35.83 Pa) and flow point (from 106.52 Pa to 269.83 Pa) compared to the wax control samples. This suggests that the addition of EC, even at half of its critical concentration, modified the structure of the wax oleogels, making them more resistant to applied shear.

#### 3.2.2. Frequency Sweep

Frequency sweep (Figure 4a–d) was performed to obtain information on the internal structure of oleogel samples, and it provides valuable evidence on the long-term stability of oleogels. All oleogel samples had higher G′ compared to G″ again, confirming the solid-like behaviour of oleogels as obtained from the amplitude sweep. The difference between the oleogels in terms of gel strength also followed the same trends as the amplitude sweep. All oleogels showed some degree of frequency-dependence behaviour, with those cooled and stored at 25 °C showing more pronounced changes as frequency increased from 0.01 to 10 Hz. Among the EC + BW oleogels stored at 25 °C (Figure 4a), the 4EC2BW had the highest G′, whereas no significant differences were found between all other treatments. However, at 4 °C (Figure 3b), the 4EC4BW that had the highest solid volume fraction (i.e., the highest total gelators content) had the highest and comparable G′ to that of the 8EC, suggesting that the cooling and storage temperature could influence the structure of oleogel, as well as its properties. On the other hand, it was difficult to observe the difference between the G′ of EC + CRW oleogels stored at 25 °C (Figure 4c), but the difference between them became more obvious when the oleogels were cooled and stored at 4 °C (Figure 4d). This finding indicates that a faster cooling stage, along with low-temperature storage, strengthens the internal structure of oleogel, producing stronger oleogels. A previous study performed on oleogels with combined wax and monoglyceride also reported more frequency-dependent behaviour when the combined gels were stored at room temperature compared to 5 °C [32]

#### 3.2.3. Oscillatory Recovery

Oscillatory recovery tests (Figure 4e–h) were performed to study the breakdown of the internal structure of oleogel samples at low and high shear rates using three-interval thixotropy tests. In the first interval (180 s), the oleogels exhibited constant and comparable G′ values to those obtained from the amplitude sweep since the shear rate was within the LVR. During the second interval (180 s), the oleogels’ structural breakdown was stimulated by increasing the shear rate from 0.1 to 10 s^−1^. When the applied shear was beyond the LVR, all oleogels showed a sudden decrease in G′. In the third interval (900 s), low shear was applied again to trigger structural regeneration. Among the control samples, the 8EC oleogel had the highest recovery (68.36%), whereas the 3BW and 6CRW samples showed very low recoveries of 1.89% and 0.88%, respectively. Similar findings were reported by [20] who observed that 3% candelilla wax/high oleic safflower oil oleogel had a recovery of 1.19%, which was attributed to the disintegration of the junction zones in the structure of the three-dimensional network structured by microplates after shear stress was applied. On the other hand, our finding contrasts with those presented in [31], where beeswax and carnauba wax/RBO oleogel had a recovery of 21.11% and 34.18%, respectively, when given 15 min to recover. A possible explanation could be that in [31]’s study, thixotropic behaviour was evaluated as a function of apparent viscosity, whereas in the current study, oscillatory recovery was examined.

The recovery (%) of the combined oleogel system (25 °C) improved compared to the wax control but was lower than that of the 8EC. The 4EC + BW oleogels had a recovery range from 14.05 to 24.50%, in the following order: 4EC2BW > 4EC1BW > 4EC3BW > 4EC4BW. The 4EC + CRW oleogels had a recovery range from 7.90 to 24.48%, in the following order: 4EC1CRW > 4EC3CRW > 4EC2CRW > 4EC4CRW. When the combined oleogel system had a poor recovery but broad yield zone (as shown in amplitude sweep), it may suggest that the oleogels are made up of a heterogeneous network of polymers and crystalline particles, leading to non-uniform bonding strength between the wax–wax and the EC–wax network [31]. Cooling and storage at 4 °C did not affect the recovery for most samples.

#### 3.2.4. Temperature Ramping

To investigate the effects of combinations of waxes and EC on the solid-to-liquid transition (i.e., when G″ is more than G′) of oleogels, temperature ramping tests were performed. This provided valuable information on their processing conditions and applications [33]. Figure 5a shows the temperature ramping graphs of selected oleogels cooled and stored at room temperature (those stored at 4 °C are not shown), and Figure 5b shows the G′G″ crossing-over temperature of all oleogels at both storage conditions (4 and 25 °C). At 25 °C, both the 3BW and 6CRW oleogels had relatively low solid–liquid transition temperatures (46.71 ± 1.5 °C and 76.96 ± 1.08 °C, respectively) compared to the combined oleogels, which had values between 97.65 and 110.13 °C. The results of wax oleogels alone were in line with the transition temperatures of the oleogel of BW (44.54 oC for 2% BW) and CRW waxes (approximately 75 oC for 5% CRW oleogel) available in the literature [33,34]. On the other hand, it is interesting to note that, over the testing range (up to 130 °C), there was no G′G″ crossing-over temperature observed for the 8EC oleogel. This is contrary to what was reported by [35], namely that that 8% EC–canola oil oleogel had a crossing-over temperature of around 130 °C. Nevertheless, the type of oil may play a role here, since RBO was used in the current study. Figure 4a shows that the G′ and G″ of the 8EC oleogel tended to cross over at approximately 107 °C (Tan δ = 0.95), but as the temperature increased, the G′ and G″ became separated apart again and exhibited a constant rubbery plateau region with G′ dominant. Nevertheless, further heating showed that the 8EC oleogel had a G″ dominant region and ultimate decomposition of polymer chains after 169.82 ± 1.2 °C (Tan δ > 1). For the combined oleogels, the solid–liquid transition temperature ranges were from 99.29 oC to 110.13 °C for 4EC + BW and 97.65 °C to 103.4 °C for 4EC + CRW oleogel. It is likely that the lower amount of EC (e.g., 4%) applied in the combined oleogels contributed to the reduction in the transition temperature compared to the 8EC control due to the diluting effect. Then, the addition of waxes may have had “plasticising” effects on EC oleogel, further reducing the solid–liquid transition temperature of the combined oleogels. Similar “plasticising” behaviour was observed with EC + stearic acid/stearyl alcohol and EC + lauric acid oleogel [18,35]. It was suggested that the small surface-active molecules cause the interruption of the polymer network, preventing the inter- and intramolecular interaction of hydrogen bonds and hindering the polymer–polymer interactions, thus reducing the temperature [35].

### 3.3. Texture Measurement

Texture measurements such as hardness, cohesiveness, adhesiveness, and gumminess are essential indicators of the possible applications of oleogels since different industrial applications require various textural needs. The texture parameters of the oleogels are shown in Figure 6a–d. 8EC had the highest hardness (122.43 ± 3.67 g) among the control samples, followed by 3BW (35.36 ± 3.58 g). It is worth noting that 6CRW (both stored at 4 and 25 °C) had a weak gel structure, and it was not able to hold its shape when the oleogel was taken out from the TPA mould. The TPA failed to detect the 6CRW correctly during testing; therefore, its data were excluded from all texture measurements. For the combined oleogel (25 °C) with EC + BW, 4EC2BW had the highest hardness (111.83 ± 1.85 g) and was also statistically insignificant compared to the 8EC. The mixed needle-like and spherulitic-shaped crystals observed under PLM (Figure 2a) might have contributed to its relatively higher hardness than other oleogel combinations. The total oleogelator concentration of 4EC2BW (i.e., 6% *w/w*) was less than the minimum concentration required for the EC only to form a self-sustained oleogel at room temperature. This suggests positive interactions between the EC polymer and beeswax crystalline particles at this specific concentration that reinforced the oleogel network, producing strong gels. Synergistic effects between EC and LMWGs in terms of mechanical properties have been reported in oleogel containing behenic acid [36], monoglycerides [17], sorbitan monostearate [37], combinations of stearyl alcohol and stearic acid [18], and lauric acid [35]. Nevertheless, positive interactions between EC and LMWGs below EC’s critical concentration have rarely been reported. Other oleogels with combined EC + BW had similar hardness values (ranging from 57.91 to 63.77 g). In terms of the EC + CRW oleogels, the 4EC3CRW had the highest hardness value (85.83 ± 4.83 g), followed by other combinations, which had statistically similar values (ranging from 54.22 to 72.33 g). In general, the oleogels cooled and stored at 4 °C had hardness levels that were higher than or comparable with those at 25 °C, with 4EC4BW (110.08 ± 7.15g) and 4EC4CRW (123.11 ± 9.72 g) producing the strongest oleogel in the combined system. Interestingly, when cooling and being stored at 25 °C, the 4EC4BW and 4EC4CRW samples did not produce the strongest oleogel among the combined systems. It is possible that the effect of higher solid volume fraction played a more significant role at lower cooling and storage temperatures, leading to enhanced interactions between EC and waxes. The large deformation test results were mostly congruent with the small deformation tests discussed above, as a direct relationship between storage modulus and hardness is expected to be observed [38].

Cohesiveness indicates how well oleogels’ internal structure is able to withstand uniaxial compressive stress. For oleogels stored at 25 °C, the 8EC oleogel had the highest cohesiveness (0.52 ± 0.04%), 3BW had the lowest among all samples (0.29 ± 0.01%), and the 4EC + wax oleogel had values in between, ranging from 0.34 to 0.4% (Figure 6b). This suggests that the 3BW had the most nonrecoverable deformation after compression but adding 4% EC to the wax system helped the oleogels recover closer to their original structure. Similar findings were observed for the oleogels stored at 4 °C, with the combined oleogels having cohesiveness values ranging from 0.350.38%, which were in between those obtained for 8EC and 3BW, and all combined oleogels showed statistically insignificant values. The gumminess index (Figure 6d) showed similar trends to the hardness index, since it was calculated by multiplying hardness and cohesiveness. Furthermore, adhesiveness is measured as the negative work between the two cycles from the force–time graph. The adhesiveness of the 4EC+BW samples (25 °C) had statistically similar values, whereas the 4EC + CRW oleogels showed an increased adhesiveness index as the wax concentration increased from 1 to 3% but decreased when 4% CRW was applied (Figure 6c). [39]’s study on carnauba wax—virgin olive oil oleogel—found that the stickiness (adhesiveness) of the oleogel increased with the carnauba wax concentration from 3 to 10% *w/w*. In our study, it is likely that the specific interactions within the 4EC4CRW oleogel (25 °C) made it less sticky compared to other treatments. It is also interesting to note that, when stored at 4 °C, the 4EC + BW oleogels all had lower adhesiveness compared to that at 25 °C, whereas the opposite was observed for the 4EC + CRW oleogels. This again confirmed that different waxes could interact differently with EC at various temperatures, producing gels with diverse properties.

### 3.4. Oil-Binding Capacity

Oil-binding capacity (OBC) is a valuable indicator of oleogels’ quality and stability. Two centrifuging speeds—2000 and 10,000 rpm—were applied to mimic different processing conditions and examine oleogels’ oil loss as a response to those conditions. In a gentle condition, 2000 rpm (Figure 6e), all oleogels had an OBC value higher than 99%, except for the 4EC1CRW, which had a value of 98.77 ± 0.3%. This indicates that the combinations of EC and BW/CRW could be beneficial in applications where gentle processing conditions are applied. On the other hand, at 25 °C and 10,000 rpm conditions (Figure 6f), the 8EC and 3BW oleogel had more than 99% oil-binding ability. This could be due to the high concentration of EC, leading to a tightly packed polymer network and the compact needle-like beeswax crystalline network binding the liquid oil [26,40]. In contrast, at the critical concentration of CRW, only 79.38 ± 2.14% OBC was observed. The microstructure analysis (Figure 2a) showed that CRW oleogel produced aggregate-like, dendritic, and unevenly distributed crystals, which are not effective for entrapping oil, causing weak conjunctions and unstable gels. This observation is concurrent with those reported by [28]. Interestingly, the OBC values of the 4EC + BW oleogels were all around 70%, but the 4EC + CRW oleogels showed a positive correlation between the wax concentration and the OBC. The OBC increased from 56% to 78% as he CRW concentration increased from 1 to 4%. This suggests that different types of waxes have various gelling mechanisms with EC. BW, which is able to gel RBO at low concentrations, had the optimal interaction, with 4EC when 2% BW was used. CRW, which required a higher concentration to gel RBO, interacted with 4EC better when 4% CRW was applied. Nevertheless, cooling and storage at 4 °C improved the OBC for all samples, with 4EC4BW and 4EC4CRW producing the highest OBC, with 94.68 ± 4.18 and 95.65 ± 0.67%, respectively. PLM (Figure 2a) showed that oleogels cooled at 4 °C had relatively smaller, more densely packed, and more evenly distributed crystals. This could lead to more surface area for the oil to adsorb onto, which could explain their higher OBC. The data show that for manufacturing oleogels for 25 °C applications, combinations of EC with BW are much more efficient than EC with CRW (1–3% CRW) for the OBC. These results are in accordance with the rheological and textural analysis.

Oscillatory recovery (tested with a rheometer using small deformation), textural properties (tested with a texture analyser and by applying large deformation), and oil binding capacity (tested using a centrifuge) studied different aspects of the oleogels. To a large extent, the findings of these techniques indicated similar trends regarding the addition of BW and CRW waxes; however, differences were seen when different levels of waxes were compared or exact property values were considered.

### 3.5. Thermal Analysis

Thermal analysis was applied to study the interactions between the EC and waxes and their contributions to oleogel properties. All oleogels went through two heating and cooling cycles, with the second cycle aiming to examine the reversibility of thermal events. The results of the first cycle are shown in Table 2 (melting stage) and Table 3 (crystallisation stage), and the thermograms of oleogels are shown in Figure 7 (oleogels cooled and stored at only 25 °C). During heating, all oleogels showed exothermic peaks in the range of 188 °C to 208 °C. This event could be related to oil thermal degradation, which was not observed in the second heating cycle. Similar findings were reported by [41] in EC–flaxseed oleogel and [25] in EC–olive–linseed–fish oil oleogel. Besides this peak, the 8EC oleogel showed no thermal transition during either the heating or the cooling stage, suggesting that its gelation mechanism and structure does not involve highly ordered secondary structure formation that can be picked up by DSC [42] Similar observations of EC oleogel were reported by [25] and [43] The 3BW and 6CRW had a T_peak_ of 47.18 ± 0.33 °C and 74.89 ± 0.23 °C, respectively. These values were lower than the T_peak_ of the raw BW (64.97 ± 1.65 °C) and CRW (83.06 ± 1.4 °C) pellets. These differences may be attributed to the presence of large quantities of oil on the crystallisation behaviour of these two waxes. Nevertheless, our result on the wax oleogel was in line with the previously reported studies showing that 2% BW in medium-chain triglycerides had a Tm peak temperature of 42.66 °C [34] and that 10% CRW in canola oil had a melting range 60 to 80 °C [44].

For the combined oleogels, peaks corresponding to the waxes’ peaks were observed; however, no peaks that could be related to EC were observed. The melting peak temperature of the combined oleogels increased significantly (*p* < 0.05) with wax concentrations. Similar findings were also reported by [45] for beeswax–sesame oil oleogel. However, there was no statistical significance in terms of the T_peak_, T_endset_, and ΔH between 3BW and 4EC3BW. This suggested that the addition of 4% EC did not influence the thermal behaviour of the 3% BW oleogel. Therefore, it seems that the increase in T_peak_ and ΔH was mainly due to the increased concentration of waxes. On the other hand, 6CRW had a T_peak_ of 74.89 ± 0.23 °C, and 4EC4CRW showed a slightly lower value of 74.53 ± 0.09 °C but was statistically insignificant with respect to that of 6CRW. However, the ΔH of 4EC4CRW (4.82 ± 0.07 J/g) was significantly lower than 6CRW (6.37 ± 1.09 J/g), highlighting the impact of wax concentration and/or the presence of oil on the alteration in the melting behaviour of CRW. Most of the melting DSC parameters of oleogels cooled and stored at 4 °C had no significant differences from those at 25 °C.

The DSC parameters of the crystallisation process (cooling stage, Figure 6b) had similar trends to those discussed above, with combined oleogels having higher T_peak_ and ΔH as the concentration of waxes increased. The crystallisation transition of BW and CRW oleogels occurred in two steps, and two crystallisation peaks were observed at 40.69 ± 0.89 °C and 25.04 ± 0.24 °C for 3BW and 56.83 ± 1.02 °C and 35.88 ± 0.71 °C for 6CRW. This confirmed the heterogeneous chemical composition of BW and CRW [34]. It is worth noting that the intensity of the first peaks (i.e., the peaks with higher temperature range) is much higher than the second Tc peak. It was explained by [34] that the first group of wax crystals formed when the T_peak-1_ temperature was reached, and then the second group of wax crystals required less energy to form the final crystal network responsible for the gelation process of wax oleogels. On the other hand, neither the 8EC nor the combined oleogels showed a crystallisation transition peak regarding EC. However, the DSC thermograms of pure EC powder showed an exothermic peak during cooling at 165.99 ± 0.61 °C. This could be because the oil acts as a plasticiser for EC, which impacts its solubility and crystallisation; hence, the ability for EC to crystallise could change, and it may only help the oleogel formation through chemical bond formation but not necessarily through crystallisation. During the oleogel cooling stage, a large amount of liquid oil (92–95%) was entrapped in the 3D polymeric network of EC that was created by inter- and intramolecular junction zones, and the presence of oil hindered the proper crystallisation of EC polymers compared to pure EC powder, and thus no obverse crystallisation transition peak was shown during analysis.

### 3.6. XRD

XRD was applied to analyse the crystalline phases and polymorphisms of the raw materials and their corresponding oleogel formulations in the 2θ range of 5–40° (Figure 8a–c). The d-spacing was calculated using Bragg’s Law. The XRD of EC showed two broad peaks at a d-spacing of 10.31 Å and 4.38 Å, and one sharp peak at 8.02 Å, which refers to the amorphous structure with small crystalline regions. The XRD pattern of 8EC oleogel showed one broad peak at 4.6 Å, which refers to the amorphous structure of EC and a low-intensity peak at 8.02 Å. A similar observation was reported by [36] in an EC–soybean-oil oleogel. BW had two sharp peaks at 4.2 Å and 3.7 Å, with the former having high relative intensity. A similar trend was observed for CRW, which had peaks located at 4.12 and 3.7 Å, which refers to the crystalline structure of the waxes. For the wax oleogel control, three distinctive peaks were revealed, including the broad peak at 4.6 Å from the RBO [18] and two sharp peaks at 4.2 Å and 3.7 Å from the waxes. The XRD pattern of 4EC + wax oleogels showed characteristics of both EC and waxes, including peaks at 8.02, 4.6, 4.2, and 3.7 Å. The addition of EC to the wax oleogel did not change the original molecular packing of the wax crystals but showed the co-existence of both EC and wax crystals. Furthermore, as expected, the intensity of peaks at 4.2 and 3.7 Å increased with the wax concentration. This highlighted that the gelator concentrations contributed to changes in the arrangement of oleogel molecules [46]. These two peaks show the characteristic of an orthorhombic subcell structure, which represents a β′ polymorphism that contributes to a smooth structure, better plasticity, and stability during storage [14]. We can observe from the findings that combining EC and wax oleogel enhanced the crystallinity and plasticity compared to the oleogel with EC alone [47,48].

### 3.7. FTIR

FTIR spectroscopy provides information on the intermolecular interactions of the oleogel network. The FTIR spectra of RBO, EC powder, BW, CRW, the control oleogels (8EC, 3BW and 6CRW), and two selected combined oleogels—4EC4BW and 4EC4CRW—are shown in Figure 8d. The broad peaks at around 3700–3600 cm ^−1^ (indicated by the red circle) were observed in the spectra of the EC powder, the 8EC oleogel, and the combined oleogels, which re linked to the stretching of -OH groups. This represents the intra- or intermolecular hydrogen bonding from EC, which contributes to the formation of oleogels with combined EC and EC through the hydrogen bonding of polymer entanglements [22]. The above-mentioned broad peak was not observed in BW or CRW, and their corresponding oleogels confirmed that the crystalline network produced by BW and CRW oleogels were based on van der Waal interactions [12]. For all oleogels, a small sharp peak was observed at 3008 cm^−1^, and two bigger sharp peaks were observed at 2922 cm^−1^ and 2853 cm^−1^, which are related to the -CH stretching from the RBO and waxes. Strong and sharp peaks at 1741 cm^−1^ link to the C=O stretching from the RBO, 1461 cm^−1^, and 1378 cm^−1^ links to the C-H bending of CH_3_ and CH_2_ groups in RBO and waxes [47]. The peaks at 1161 cm^−1^ and 1094 cm^−1^ represent the C-O stretching of C-O-C and C-O-H from the RBO and waxes. The sharp peaks at 718 cm^−1^ confirmed the presence of (CH2)_n_ bending from RBO [22,47]. All combined oleogels had similar FTIR spectra, despite the different concentration of waxes in the system and the storage temperatures. The FTIR spectra of 8EC oleogel and the EC powder show clear differences at ~1500 to 750 cm^−1^, which may indicate the different molecular conformations for EC in its powder form and EC in the oleogel formulation.

## 4. Conclusions

The current study systemically investigated the interactions between EC and two types of waxes, BW and CRW, to structure RBO. The high concentration of EC required to create oleogels has been a major hurdle in its commercial application. Our study found that with the addition of as low as 1% *w/w* BW/CRW, EC is able to form self-sustaining oleogels at room temperature when only 50% of its critical concentration is applied. Combined EC-Wax oleogels would help address EC concerns relating to high costs and the consumption of high levels of EC as a chemically modified hydrocolloid. The addition of 4% *w/w* BW/CRW to 4% *w/w* EC (cooled and stored at 4 °C) produced the optimised sample in terms of rheological and mechanical properties and the OBC compared to all other treatments, and it had comparable values to those of 8EC, as well as improved plasticity. We also demonstrated that different types of waxes have different structure-forming capabilities and interact differently with EC. This led to different impacts on the rheological and microscopic structures of BW and CRW, producing oleogels with various properties. Microscopic analysis showed considerable changes in crystal morphology, which formed smaller wax crystals upon EC addition. This study demonstrated the synergistic interaction between EC and waxes with a carefully designed system and highlighted this system’s unique potential in developing application-specific oleogels. Future studies can focus on the in-depth study of the microstructure of EC and wax oleogels using different microscopic techniques to observe the surface morphology, which is beneficial for further understanding the oil-structuring ability of the combined oleogels. Significant differences observed among the EC oleogels when BW and CRW are incorporated in terms of crystallinity level, solid-to-liquid transition temperature, and also microscopic appearance may indicate organoleptic differences and require further study. Secondary treatments during sample preparation, such as sonication and annealing, could also have the potential to improve the functional properties of the combined oleogels. Our study also showed that new gelators that can help create oleogels with low concentrations and simpler preparation procedures are of industrial interest and are worth investigating.

## Figures and Tables

**Figure 1 foods-12-03093-f001:**
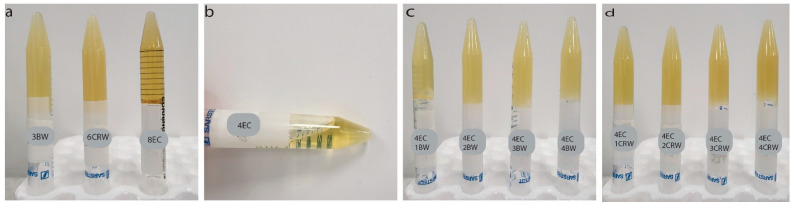
Visual appearance of oleogels. Visual appearance of oleogels after preparation and storage at 25 °C (**a**) control oleogels (3BW, 6CRW, and 8EC); (**b**) 4EC oleogel; (**c**) 4EC + BW (i.e., 1BW, 2BW, 3BW, and 4BW) oleogels; (**d**) 4EC + CRW (i.e., 1CRW, 2CRW, 3CRW, and 4CRW) oleogels. BW: beeswax; CRW: carnauba wax EC: ethylcellulose; the prefix numbers (1, 2, 3, 4, 6, and 8) denote the oleogelator concentrations (*w/w*) used in the control and multiple component oleogels.

**Figure 2 foods-12-03093-f002:**
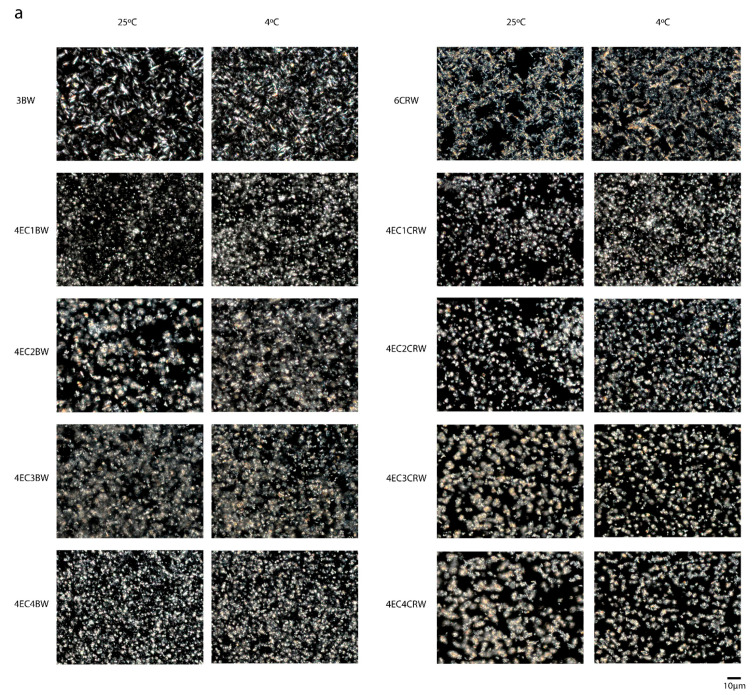
Microstructures of oleogel. (**a**) PLM images of oleogel that was cooled and stored at 25 °C and 4 °C; scale is 10 µm. (**b**) CLSM images of selected oleogels that were cooled and stored at 25 °C; scale is 50 µm. The reddish-white particles are indicated using white arrows indicating EC particles.

**Figure 3 foods-12-03093-f003:**
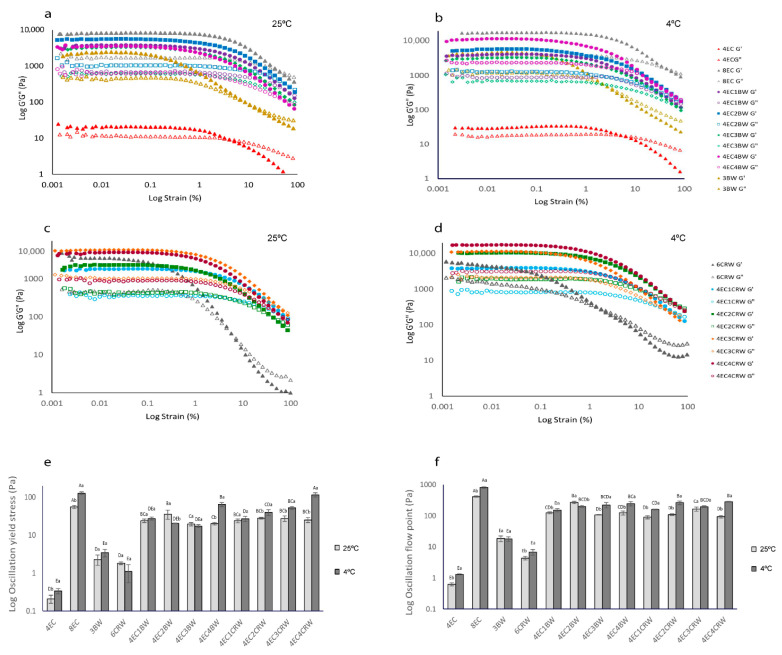
Amplitude sweep of oleogels, yield stress, and flow point of oleogels. Strain sweep of oleogels. 4EC, 8EC, 3BW, and 4EC + BW oleogel (**a**) cooled and stored at 25 °C (**b**) cooled and stored at 4 °C; 6CRW and 4EC + CRW oleogel (**c**) cooled and stored at 25 °C; (**d**) cooled and stored at 4 °C; (**e**) yield stress; (**f**) flow point of oleogels that were cooled and stored at 25 °C (shown in light grey) and 4 °C (shown in dark grey). Different upper-case letters indicate the significant difference (*p* < 0.05) of different oleogel treatments at the same storage temperature, and different lowercase letters indicate the significant difference (*p* < 0.05) among the same oleogel treatments at different storage temperatures.

**Figure 4 foods-12-03093-f004:**
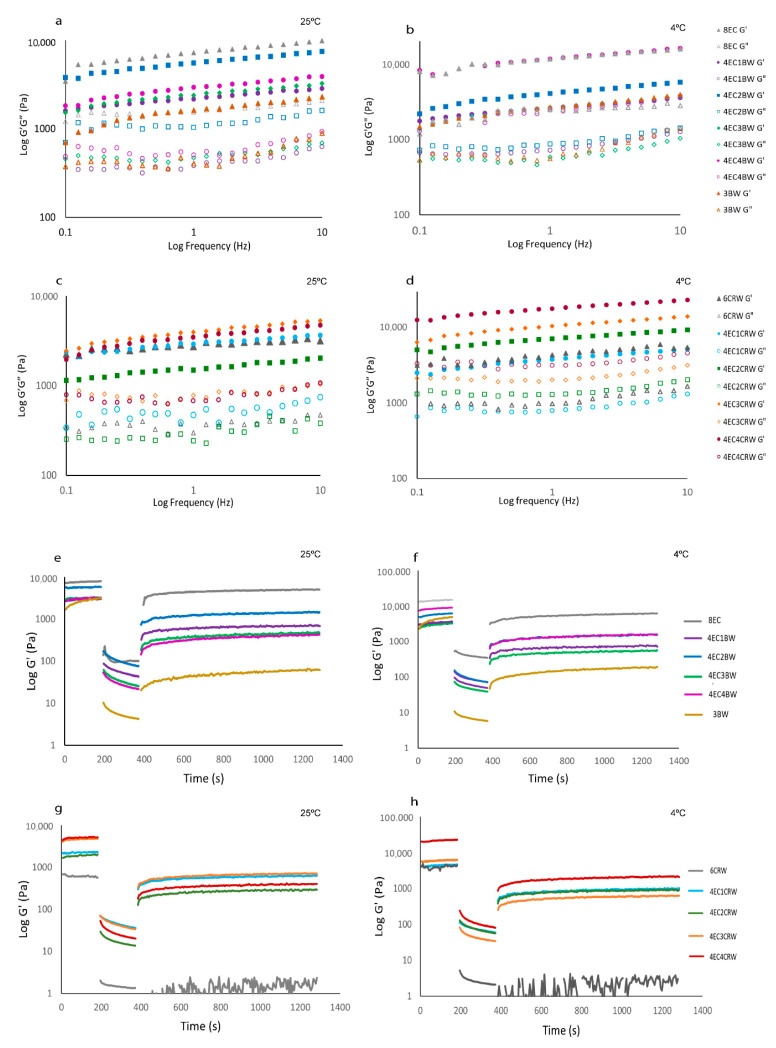
Frequency sweep and oscillation recovery of oleogels. Frequency sweep of oleogels. 8EC, 3BW, and 4EC + BW oleogel (**a**) cooled and stored at 25 °C; (**b**) cooled and stored at 4 °C. 6CRW and 4EC + CRW oleogel (**c**) cooled and stored at 25 °C; (**d**) cooled and stored at 4 °C. Oscillation recovery of oleogels. 8EC, 3BW and 4EC + BW oleogel (**e**): cooled and stored at 25 °C; (**f**): cooled and stored at 4 °C. 6CRW and 4EC + CRW oleogel (**g**): cooled and stored at 25 °C; (**h**): cooled and stored at 4 °C.

**Figure 5 foods-12-03093-f005:**
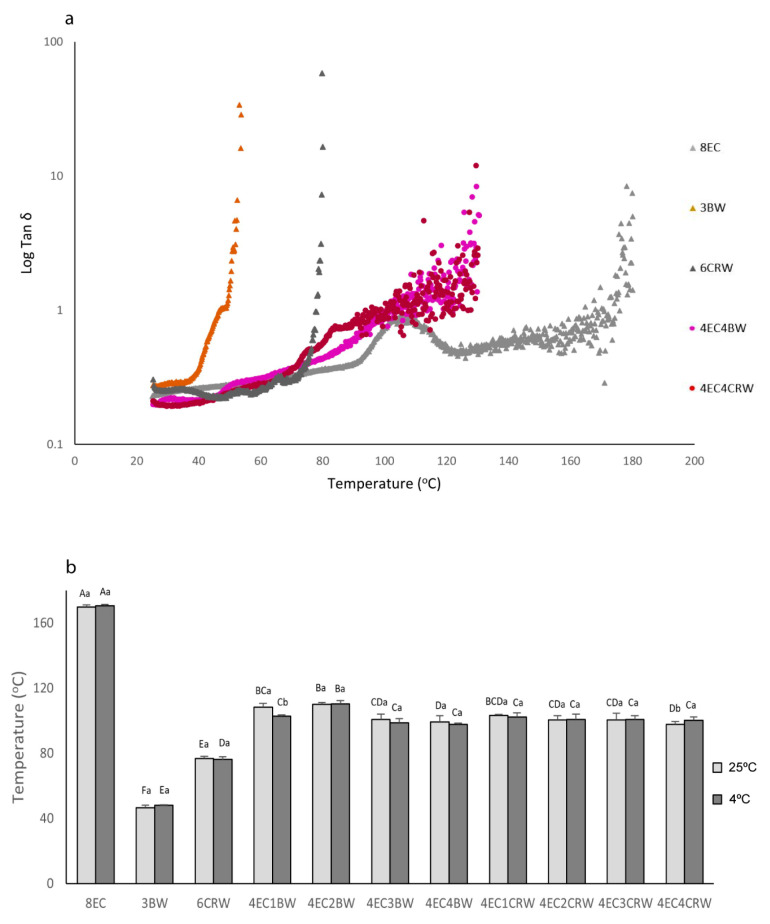
Temperature ramp and G′G″ crossing-over temperature of oleogels. (**a**) Temperature ramp graphs of selected oleogel samples indicating a change of tan δ versus temperature; (**b**) G′G″ crossing-over temperature of oleogels that were cooled down and stored at 25 °C (shown in light grey) or 4 °C (shown in dark grey). Different lowercase letters above each column indicate significant differences (*p* < 0.05) between different storage temperatures in the same group. Different uppercase letters indicate significant differences (*p* < 0.05) between the same storage temperature of different oleogel treatments, and different lowercase letters indicate significant differences (*p* < 0.05) between different storage temperatures among the same oleogel treatments.

**Figure 6 foods-12-03093-f006:**
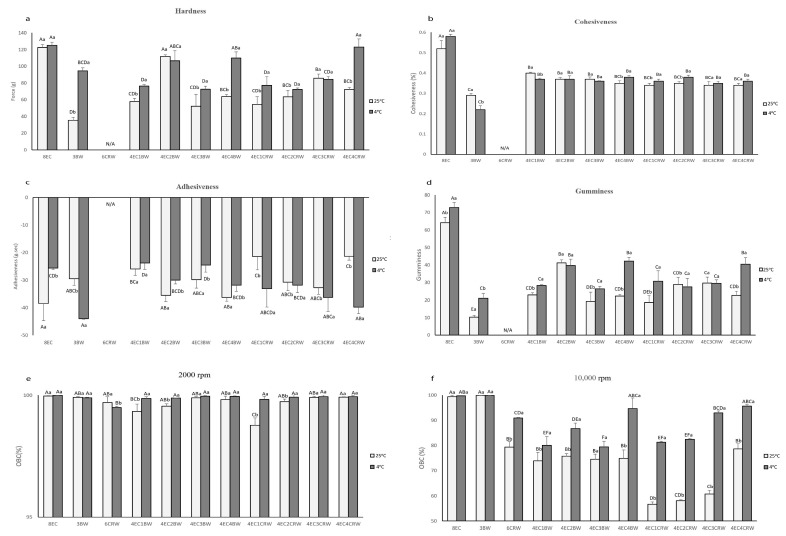
Texture parameters and OBC of oleogels. Texture parameters of oleogels cooled and stored at 25 °C (shown in light grey) and 4 °C (shown in dark grey): (**a**) hardness; (**b**) cohesiveness; (**c**) adhesiveness; (**d**) gumminess. Oil-binding capacity (OBC) of oleogels that were cooled and stored at 25 °C (shown in light grey) and 4 °C (shown in dark grey); (**e**) centrifuging speed at 2000 rpm and (**f**) 10,000 rpm. Different lowercase letters above each value indicate significant differences (*p* < 0.05) between different storage temperatures in the same group. Different uppercase letters indicate significant differences (*p* < 0.05) between the same storage temperature of different oleogel treatments, and different lowercase letters indicate significant differences (*p* < 0.05) between different storage temperatures among the same oleogel treatments.

**Figure 7 foods-12-03093-f007:**
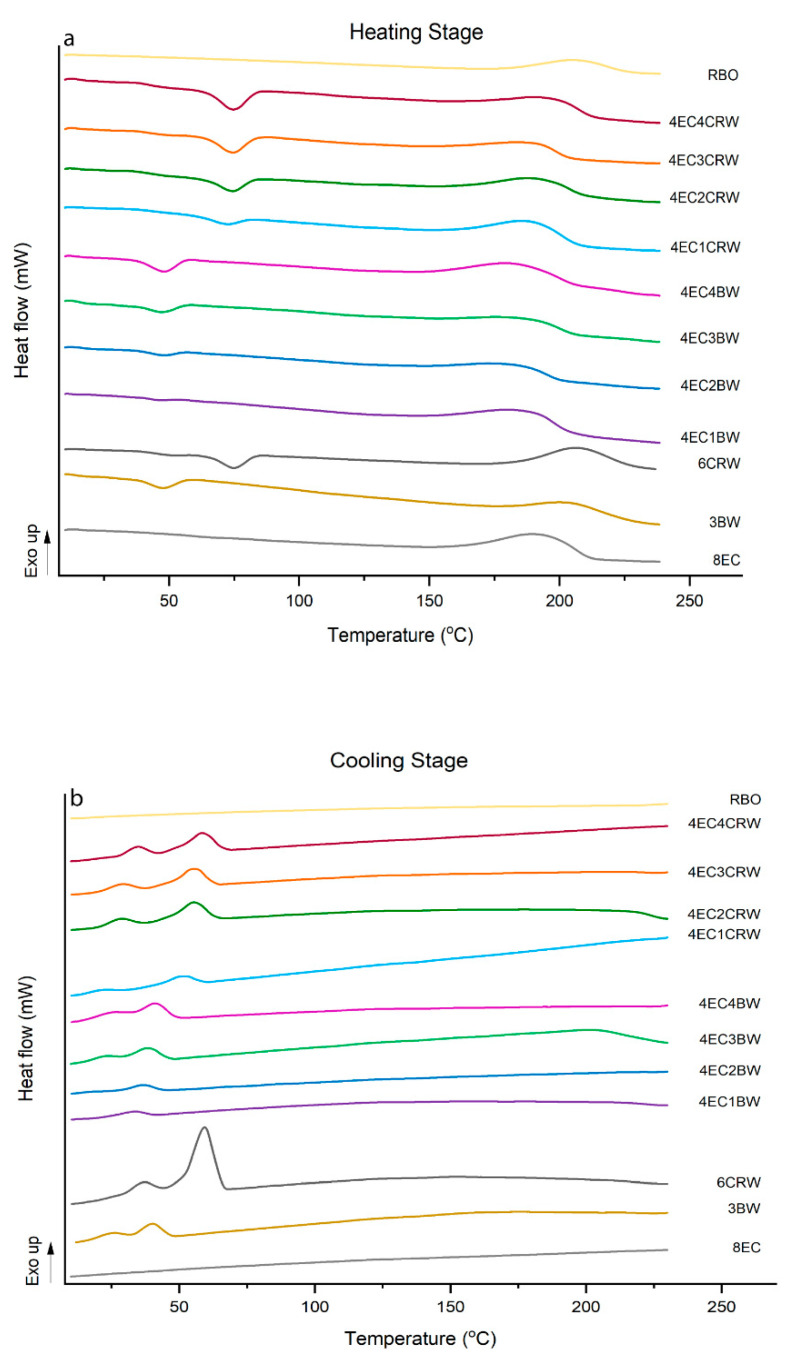
Thermograms of oleogels. Differential scanning calorimetry (DSC) thermograms of oleogels that were cooled down and stored at 25 °C; (**a**) heating stage and (**b**) cooling stage.

**Figure 8 foods-12-03093-f008:**
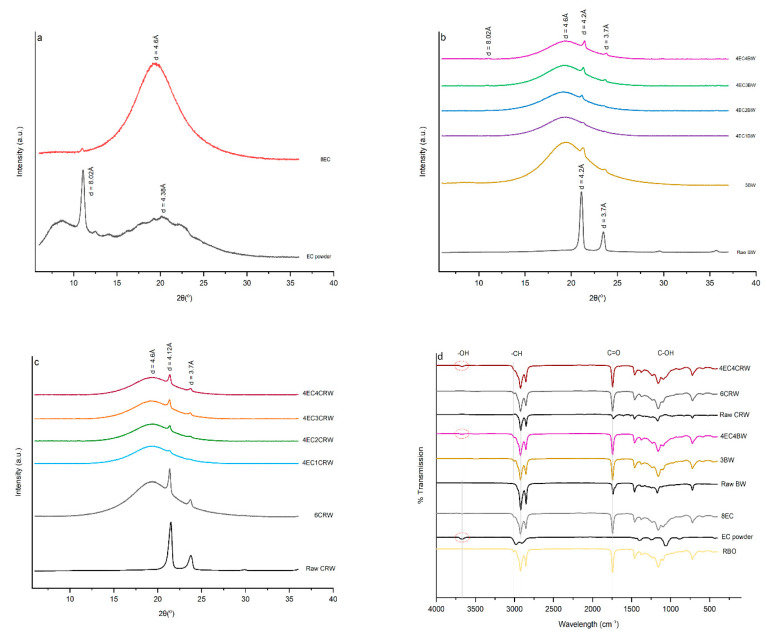
XRD pattern and FTIR spectrum of oleogels. X-ray diffraction (XRD) patterns of (**a**) EC powder and 8EC oleogel; (**b**) raw BW and EC + BW oleogels that cooled and stored at 25 °C; (**c**) raw CRW and EC + CRW oleogels that cooled and stored at 25 °C. Fourier transforms infrared (FTIR) spectra of (**d**) raw materials (RBO, EC powder, raw BW, and raw CRW) and selected oleogels that were cooled down and stored at 25 °C.

**Table 1 foods-12-03093-t001:** Formulation of the control oleogel and multicomponent oleogels.

Sample Code	EC (*w/w*)	Beeswax (*w/w*)	Carnauba Wax (*w/w*)	Oil (*w/w*)
Controls				
4EC	4	0	0	96
8EC	8	0	0	92
3BW	3	0	0	97
6CRW	6	0	0	94
Treatments				
4EC1BW	4	1	0	95
4EC2BW	4	2	0	94
4EC3BW	4	3	0	93
4EC4BW	4	4	0	92
4EC1CRW	4	0	1	95
4EC2CRW	4	0	2	94
4EC3CRW	4	0	3	93
4EC4CRW	4	0	4	92

**Table 2 foods-12-03093-t002:** The melting behaviour indicators of oleogels. Different lowercase letters in each column indicate significant differences (*p* < 0.05) between each parameter of 4 °C and 25 °C oleogels. Different uppercase letters indicate significant differences (*p* < 0.05) among all treatments presented in each row. The “-” indicated no thermal behaviour was dectected in the testing temperature.

DSC Parameters	8EC	3BW	6CRW	4EC + BW				4EC + CRW			
				4EC1BW	4EC2BW	4EC3BW	4EC4BW	4EC1CRW	4EC2CRW	4EC3CRW	4EC4CRW
25 °C											
Melting											
Tm—onset (°C)	-	38.77 ± 0.37 DEa	65.22 ± 0.13 Aa	39.45 ± 1.2 Da	39.34 ± 0.67 DEa	36.73 ± 0.28 Fb	38.35 ± 0.16 Ea	61.05 ± 0.3 Ca	64.05 ± 0.27 Ba	64.35 ± 0.14 ABa	64.54 ± 0.13 ABa
Tm—peak (°C)	-	47.18 ± 0.33 Da	74.89 ± 0.23 Aa	46.09 ± 0.37 Ea	47.56 ± 0.56 Da	47.64 ± 0.29 Da	48.98 ± 0.68 Ca	71.86 ± 0.2 Ba	74.2 ± 0.16 Aa	74.33 ± 0.17 Aa	74.53 ± 0.09 Aa
Tm—endset (°C)	-	58.87 ± 0.45 Ca	85.32 ± 0.11 Bb	53.58 ± 1.55 Ea	55.73 ± 0.87 Db	59.76 ± 0.14 Ca	59.16 ± 1.38 Cb	83.45 ± 1.21 Ba	85.13 ± 0.84 Ba	87.63 ± 1.05 Aa	87.54 ± 0.28 Aa
ΔH (J/g)	-	1.74 ± 0.15 Ea	6.37 ± 1.09 Aa	0.25 ± 0.12 Ga	0.75 ± 0.08 FGb	1.74 ± 0.14 Ea	2.63 ± 0.09 Da	1.44 ± 0.08 EFa	2.79 ± 0.38 CDa	3.58 ± 0.16 Ca	4.82 ± 0.07 Ba
4 °C											
Tm—onset (°C)	-	36.99 ± 0.22 Db	64.25 ± 0.08 Ab	32.15 ± 2.3 Eb	38.28 ± 0.62 CDa	39.07 ± 0.51 Ca	37.93 ± 0.08 CDb	61.49 ± 0.5 Ba	64.5 ± 0.23 Aa	63.92 ± 0.61 Aa	64.58 ± 0.06 Aa
Tm—peak (°C)	-	45.76 ± 0.36 Eb	74.39 ± 0.19 Ab	40.05 ± 1.2 Fb	46.33 ± 0.1 Eb	47.82 ± 0.39 Da	49.61 ± 0.17 Ca	71.24 ± 0.4 Bb	74.3 ± 0.19 Aa	73.75 ± 0.18 Ab	74.12 ± 0.06 Ab
Tm—endset (°C)	-	57.6 ± 0.91 Da	86.47 ± 0.37 Aa	49.46 ± 0.41 Eb	57.98 ± 0.29 Da	59.03 ± 0.59 Da	61.55 ± 0.3 Ca	81.05 ± 0.3 Bb	86.71 ± 1.07 Aa	87.55 ± 1.07 Aa	87.34 ± 0.59 Aa
ΔH (J/g)	-	1.56 ± 0.21 Eb	7.04 ± 0.41 Aa	0.24 ± 0.06 Ga	0.95 ± 0.04 Fa	1.53 ± 0.07 Ea	2.26 ± 0.09 Db	1.01 ± 0.04 Fb	2.15 ± 0.11 Db	3.55 ± 0.21 Ca	4.31 ± 0.31 Bb

**Table 3 foods-12-03093-t003:** The crystallisation behaviour indicators of oleogels. Different lowercase letters in each column indicate significant differences (*p* < 0.05) between each parameter of 4 °C and 25 °C oleogels. Different uppercase letters indicate the significant differences (*p* < 0.05) among all treatments presented in each row. The “-” indicated no thermal behaviour was dectected in the testing temperature.

DSC Parameters	8EC	3BW	6CRW	4EC + BW				4EC + CRW			
				4EC1BW	4EC2BW	4EC3BW	4EC4BW	4EC1CRW	4EC2CRW	4EC3CRW	4EC4CRW
25 °C											
Crystallisation											
Tc—onset (°C)	-	47.52 ± 0.75 EFa	64.76 ± 1.37 ABa	39.36 ± 0.94 Ha	43.26 ± 0.15 Ga	46 ± 0.66 Fa	48.35 ± 1.65 Ea	58.62 ± 0.2 Da	62.34 ± 0.74 Ca	63.56± 0.79 BCa	66.2 ± 0.12 Aa
Tc—peak 1 (°C)	-	40.69 ± 0.89 Ea	56.83 ± 1.02 Bb	32.76 ± 0.51 Ha	36.57 ± 0.06 Ga	38.14 ± 0.12 Fa	40.78 ± 0.61 Ea	51.23 ± 0.5 Da	54.49 ± 0.57 Ca	56.63 ± 1.02 Ba	58.21 ± 0.08 Aa
Tc—peak 2 (°C)	-	25.04 ± 0.24 Da	35.88 ± 0.71 Ab	-	19.41 ± 1.29 Fa	22.8 ± 0.33 Ea	25.32 ± 0.44 Da	22.95 ± 0.67 Ea	27.44 ± 0.99 Ca	30.09 ± 0.94 Ba	34.57 ± 0.09 Aa
Tc—endset (°C)	-	14.65 ± 0.38 Eb	18.47 ± 1.19 Cb	19.48 ± 3.09 BCa	13.03 ± 0.74 Eb	14.53 ± 0.08 Eb	14.24 ± 0.86 Ea	15.13 ± 0.33 DEa	17.59 ± 0.18 CDa	21.37 ± 1.4 Ba	25.86 ± 0.32 Aa
ΔH (J/g)	-	2.63 ± 0.1 Ea	9.47 ± 0.85 Aa	0.5 ± 0.13 Ga	1.34 ± 0.04 Fa	2.5 ± 0.12 Ea	3.61 ± 0.19 Da	1.59 ± 0.12 Fa	3.59 ± 0.39 Da	4.57 ± 0.39 Ca	5.98 ± 0.05 Ba
4 °C											
Tc—onset (°C)	-	47.09 ± 0.32 Ea	66.83 ± 1.9 Aa	39.75 ± 0.42 Ha	43.73 ± 0.38 Ga	45.47 ± 0.34 Fa	47.41 ± 0.22 Ea	57.67 ± 0.36 Db	61.35 ± 0.53 Ca	64.67 ± 0.25 Ba	65.57 ± 0.39 ABa
Tc—peak 1 (°C)	-	40 ± 0.54 Ea	60.1 ± 2.11 Aa	33.29 ± 0.51 Ha	36.96 ± 0.54 Ga	38.04 ± 0.16 FGa	39.32 ± 0.29 EFb	50.71 ± 0.17 Da	54.16 ± 0.43 Ca	57.78 ± 0.33 Ba	58.15 ± 0.57 Ba
Tc—peak 2 (°C)	-	25.24 ± 0.04 Da	38.05 ± 1.01 Aa	-	19.73 ± 0.56 Fa	22.38 ± 0.81 Ea	25.12 ± 0.71 Da	21.64 ± 0.52 Ea	26.78 ± 1.02 Da	31.63 ± 1.36 Ca	33.67 ± 0.31 Bb
Tc—endset (°C)	-	15.77 ± 0.4 Fa	27.91 ± 1.05 Aa	20.37 ± 0.35 Da	14.74 ± 0.49 Fa	14.78 ± 0.09 Fa	16 ± 0.7 Fa	15.81 ± 0.87 Fa	18.11 ± 0.89 Ea	23.02 ± 1.33 Ca	26.34 ± 0.24 Ba
ΔH (J/g)	-	2.62 ± 0.13 Ea	8.56 ± 0.34 Aa	0.55 ± 0.06 Ga	1.44 ± 0.07 Fa	2.15 ± 0.22 Ea	3.33 ± 0.15 Da	1.43 ± 0.07 Fa	2.68 ± 0.49 DEa	5.03 ± 0.36 Ca	5.77 ± 0.68 Ba

## Data Availability

The data presented in this study are available onrequest from the corresponding author.

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
