# Peer review of "Multicomponent Oleogels Prepared with High- and Low-Molecular-Weight Oleogelators: Ethylcellulose and Waxes"

_foods, 2023, doi:10.3390/foods12163093_

Round 1
Reviewer 1 Report
The paper “Multicomponents oleogeld prepared with high and low molecular weight oleogelators: ethylcellulose and waxes” describes the properties of 12 selected oleogel samples prepared with rice bran oil (RBO) as a liquid oil and three different additives – the polymer ethylcellulose (EC) and two waxes – beeswax (BW) and carnauba wax (CBW). After preparing the selected samples, the authors have examined in details their properties, incl. macroscopic and microscopic appearance, rheological and texture properties, thermal behavior and structural characteristics. The results from all these experiments are presented in a systematic way in the manuscript.
Although an extensive experimental work has been carried out, the authors only present and describe the obtained results. There are no general conclusions for the effect of one or another examined factor which may be valuable for the readers of the paper. Instead of this, there are several places within the manuscript, incl. in the conclusions section where the authors state that “We also demonstrated that the different types of waxes interact differently with EC, producing oleogels with various properties”. In my opinion, the aim of such paper would be to extract some guiding rules about the observed effects which may explain what are the observed differences caused by and how the different properties are explained with respect to the composition, otherwise – the study is a simple (very long) description of a numerous experimental results.
Another main conclusion stated in the Conclusion section: “The addition of 4% w w-1…. Produced the optimized sample in terms of rheological, mechanical properties …” is also not entirely supported by the results presented in the paper, as there are several types of measurements in which the authors show that the sample 4EC2BW for example have better properties that 4EC4BW.
The following points should be also considered by the authors:
- The authors should explain what they mean by the term “interaction” which is used numerous times in the paper when the synergistic properties of EC and waxes are mentioned, incl. in the first sentence of the Conclusions section. Usually the term is used as a reference to VdW or electromagnetic forces, covalent or H-bonds, etc. However, no indications for presence of such interactions are presented in the manuscript, and even more – the FTIR data for example, shows that in fact – there are no such interactions. Maybe – revision of the use of the term is required.
- Critical concentration – the authors mentioned numerous times that they work at the critical concentrations of the respective gelators (incl. in the abstract and in the conclusions). However, this claim is not really supported by the included experimental results. The only sample for which two different concentrations of the gelators are tested (giving different final appearance of the samples) are 4EC and 8EC. It is unclear what will be the behavior of 5EC, 6EC or 7EC samples. This also applies for the critical concentration of the waxes. Furthermore, lines 311-312 of the paper state that 1-1.5 wt. % BW has been previously found to be sufficient to produce oleogel with RBO, whereas in the present paper 3 wt.% BW is used and claimed as being at the critical concentration.
This point is important also to understand the contribution of wax and EC for the combined properties of the mixed samples, i.e. if 1 or 2% BW is sufficient to produce oleogel – do we really need to include also the 4EC in the oleogels or lower concentrations would give similar results?
These control experiments should be included to allow better understanding of the optimal composition.
- There are several sections within the paper, where the obtained results show non-monotonous behavior with respect to the wax content, see for example 3.2.3, 3.3, 3.4... The reason for this should be explained and confirmed with appropriate model experiments.
- Melting temperature of the oleogels, section 3.2.4 – this term should be possibly revised, as by melting one usually refers to the solid-to-liquid phase transition. In this case, however, the authors refer to the melting of the included wax and not of the whole oleogel, which contains predominantly liquid oil.
- The structuring mechanism of the EC should be explained in the introduction. Is there any reason why one should expect to see peaks in the DSC thermogram of EC samples? The discussion about this can be removed.
- The introduction section could be improved by explaining the main methods for production of oleogels and the results which are available with similar systems.
- The chemical composition of the substances used should be included.
- RBO stability at high temperature – has the authors checked whether the preparation procedure affects the unsaturated fatty acid residues included in the RBO triglyceride molecules? Indications for such effects are mentioned by the obtained DSC thermograms.
- DSC results from the samples stored at 4 and 25 degs. What is the reason for the observed differences with respect to the melting enthalpies for the same sample stored at 4 and 25 deg? For example 4EC2CRW has enthalpy of 2.79 ± 0.38 J/g when stored at 25 deg and 2.15 ± 0.11 J/g when stored at 4 deg. As the sample are claimed to have the exact same composition – one would expect the same enthalpy or possible reversed behavior (if further crystallization takes place at lower T). This question also applies to many of the other results. Also, what is the reason for significantly higher crystallization enthalpies?
- XRD part – crystals of EC are mentioned – the DSC does not support this claim, should be revised. “Enhanced crystallinity” claim should be revised.
- Second p.2 in the XRD section – the sentence citing Yang et al. 2017 should be revised.
- XRD – the spectra of pure RBO should be also included.
Another minor comments:
- Line 29 “plastic fats” – what do you mean by this?
- Lines 38-39 – references should be added
- Lines 247-248 – how the stated cooling rates have been measured? Is the cooling rate linear throughout the whole time from the initial high T down to the low desired T?
- Lines 240-253 – should be revised to state clearly what are the credible findings from these experiments. Pictures shown in Fig. 2a can be zoomed in to show more clearly the discussed structures.
- Lines 466-467 – the trends shown in Fig. 6b and 6d are not really “similar”.
- Lines 483-484 and 516-517 – these sentences should be revised.
- Discussion about DSC results on the second p.1 – should be revised.
- First sentence on the second p. 4 (above the conclusion section) – should be revised.
- The font of the “G” symbols within the manuscript should be unified.
- Figure 3b – there are two curves with full pink circles.
- Line 500 “more efficient that” -> than
- Second p.1 (DSC part) “Statically” -> statistically
Author Response
Response to Reviewers’ comments:
Multicomponent oleogels prepared with high and low molecular weight oleogelators: ethylcellulose and waxes
Ziyu Wang, Jayani Chandrapala, Tuyen Truong & Asgar Farahnaky
We would like to thank all the reviewers for their valuable time and providing these constructive comments. We have addressed all and revised the manuscript.
The line numbers referred to in this document are the lines of the revised manuscript.
Reviewer 1:
- Although an extensive experimental work has been carried out, the authors only present and describe the obtained results. There are no general conclusions for the effect of one or another examined factor which may be valuable for the readers of the paper. Instead of this, there are several places within the manuscript, incl. in the conclusions section where the authors state that “We also demonstrated that the different types of waxes interact differently with EC, producing oleogels with various properties”. In my opinion, the aim of such paper would be to extract some guiding rules about the observed effects which may explain what are the observed differences caused by and how the different properties are explained with respect to the composition, otherwise – the study is a simple (very long) description of a numerous experimental results.
Answer: Conclusion section has been revised, please see the revised version.
“As highlighted, the high concentration of EC required to create oleogels has been a major hurdle against its commercial application. Our study found that with the addition of as low as 1% w w-1 BW/CRW, EC is able to form self-sustained oleogels at room temperature when only 50% of its critical concentration is applied.”
- Another main conclusion stated in the Conclusion section: “The addition of 4% w w-1…. Produced the optimized sample in terms of rheological, mechanical properties …” is also not entirely supported by the results presented in the paper, as there are several types of measurements in which the authors show that the sample 4EC2BW for example have better properties that 4EC4BW.
Answer: The best oleogels results not only depended on the % of the gelaotor but also the storage (incubation) temperature. It has been mentioned in the manuscript that 4EC2BW have better properties than 4EC4BW at room temperature but overall, 4EC4BW and 4EC4CRW cooled and stored at 4°C produced the best properties, as higher level of crystallization was created. E.g., please see the 1st paragraph of the Conclusion section for “The addition of 4% w w-1 BW/CRW to 4% w w-1 EC (cooled and stored at 4℃) produced the optimized sample in terms of rheological, mechanical properties, and OBC compared to all other treatments and had comparable values to that of the 8EC as well as improved plasticity.”
- The authors should explain what they mean by the term “interaction”which is used numerous times in the paper when the synergistic properties of EC and waxes are mentioned, incl. in the first sentence of the Conclusions section. Usually the term is used as a reference to VdW or electromagnetic forces, covalent or H-bonds, etc. However, no indications for presence of such interactions are presented in the manuscript, and even more – the FTIR data for example, shows that in fact – there are no such interactions. Maybe – revision of the use of the term is required.
Answer: thank you for the comment. We have clarified this in the conclusion section, also please see the below in the Conclusion section:
“We also demonstrated that different types of waxes have different structure forming capabilities and interact differently with EC. This led to different impacts on rheological and microscopic structures of BW and CRW, producing oleogels with various properties.”
- Critical concentration – the authors mentioned numerous times that they work at the critical concentrations of the respective gelators (incl. in the abstract and in the conclusions). However, this claim is not really supported by the included experimental results. The only sample for which two different concentrations of the gelators are tested (giving different final appearance of the samples) are 4EC and 8EC. It is unclear what will be the behavior of 5EC, 6EC or 7EC samples. This also applies for the critical concentration of the waxes. Furthermore, lines 311-312 of the paper state that 1-1.5 wt. % BW has been previously found to be sufficient to produce oleogel with RBO, whereas in the present paper 3 wt.% BW is used and claimed as being at the critical concentration.
Answer: For the wax containing oleogels, critical concentration waxes depend on the wax type (BW or CRW), temperature (both preparation temperature and storage temperature) and oil type. For EC in addition to oil type and oleogel preparation/storage temperatures, the molecular weight and SD (substitution degree) are impacting critical concentration. Therefore prior to conducting the main experimental work, preliminary work was performed, and these critical concentrations were found and reported along with these conditions.
The following paragraph was revised/added to the 2.2 section (Preparation of oleogels) to address this comment:
“Generally, the critical concentration of oleogelators depends on a number of factors including oil type, oleogel type, preparation method and storage temperature and time. Since each gelator has different gelling mechanisms and capacities, the critical concentration of each gelator at room temperature (8%, 3%, and 6% w w-1, respectively) is selected as the control concentration instead of choosing a constant concentration for all gelators. These critical concentrations were obtained by preparing a series of olegoles using a range of concentrations of each of these oleogelators, and testing their ability to create oleogels after overnight storage.”
- There are several sections within the paper, where the obtained results show non-monotonous behavior with respect to the wax content, see for example 3.2.3, 3.3, 3.4... The reason for this should be explained and confirmed with appropriate model experiments.
Answer: section 3.2.3 refers to oscillatory recovery test using small deformation testing, section 3.3 refers to Textural properties (measured using large deformation technique) and section 3.4 discusses oil binding capacity measured using centrifugal forces. It is evident that the oleogels studied respond differently to different stresses (large, small, centrifugal) and therefore, the same patterns have not been seen for all these 3 techniques. This sentence has been added to the paper to highlight this fact. Please see the end of Section 3.4 and below.
“Oscillatory recovery (tested with a rheometer using small deformation), textural properties (tested with a Texture analyser and by applying large deformation) and oil binding capacity (tested using a centrifuge) studied different aspects of the oleogels. To a alge extend, the findings of these techniques indicated similar trends on the addition of BW and CRW waxes, however, differences were seen when different levels of waxes are compared or exact property values are considered.”
- Melting temperature of the oleogels, section 3.2.4 – this term should be possibly revised, as by melting one usually refers to the solid-to-liquid phase transition. In this case, however, the authors refer to the melting of the included wax and not of the whole oleogel, which contains predominantly liquid oil.
Answer: thank you, this has been taken into account and melting point has been replaced with “solid to liquid transition” across the Section 3.2.4 as below: “To investigate the effects of combinations of waxes and EC on the solid to liquid transition (i.e., when G′′ is more than G′) = of oleogels, temperature ramp tests were performed. This provides valuable information on their processing conditions and applications (Lim, Jeong, et al., 2017). Figure 5a shows the temperature ramp graphs of selected oleogels cooled and stored at room temperature (those stored at 4oC were not shown), and Figure 5b shows the G′G′′ crossing-over temperature of all oleogels at both storage conditions (4 and 25℃). At 25oC, both the 3BW and 6CRW oleogels had a relatively low solid-liquid transition temperatures (46.71 ± 1.5oC and 76.96 ± 1.08oC, respectively) compared to the combined oleogels which had values between 97.65 and 110.13oC. The results of wax oleogels alone were in line with the transition temperatures of the oleogel of BW (44.54oC for 2% BW) and CRW waxes (approximately 75oC for 5% CRW oleogel) available in the literature (Lim, Jeong, et al., 2017; Martins et al., 2016). On the other hand, it is interesting to note that, over the testing range (up to 130oC), there was no G′G′′ crossing-over temperature observed for the 8EC oleogel. This is contrary to what was reported by Haj Eisa et al. (2020) that 8% EC – canola oil oleogel had a crossing-over temperature of around 130oC. Nevertheless, the type of oil may play a role here since RBO was used in the current study. Figure 4a presents that the G′ and G′′ of the 8EC oleogel tend to cross over at approximately 107oC (Tan δ = 0.95), but as the temperature increases, the G′ and G′′ became separated apart again and exhibited a constant rubbery plateau region with G′ dominant. Nevertheless, further heating showed that the 8EC oleogel had a G′′ dominant region and ultimate decomposition of polymer chains after 169.82 ± 1.2oC (Tan δ > 1). For the combined oleogels, the solid-liquid transition temperature ranges from 99.29oC to 110.13oC for 4EC + BW and 97.65oC to 103.4oC for 4EC + CRW oleogel. It is likely that the lower amount of EC (e.g., 4%) applied in the combined oleogels contributed to the the reduction in the transition temperature compared to the 8EC control due to the diluting effect. Then the addition of waxes may had “plasticizing” effects on EC oleogel, further reduce the solid-liquid transition temperature of the combined oleogels. Similar “plasticizing” behavior was observed with EC + stearic acid/stearyl alcohol and EC + lauric acid oleogel (Gravelle et al., 2017; Haj Eisa et al., 2020). It was suggested that the small surface-active molecules cause interruption of the polymer network, preventing the inter-and intramolecular interaction of hydrogen bonds and hindering the polymer-polymer interactions, thus reducing the temperature (Haj Eisa et al., 2020).”
- The structuring mechanism of the EC should be explained in the introduction. Is there any reason why one should expect to see peaks in the DSC thermogram of EC samples?
Answer: The structuring mechanism of EC has now added to the Introduction section in line 57-61. As below” When EC is heated above its glass transition temperature, the strands of EC became flexible and rubbery, allow their interaction with solvents, leads to dissolution; upon cooling, the liquid oil is physically entrapped within the 3D polymer network as a result of the formation of inter- and intramolecular junction zones among EC macromolecules (Laredo et al., 2011). “
- The introduction section could be improved by explaining the main methods for production of oleogels and the results which are available with similar systems.
Answer: to address this, we have added a sentence and referred the readers to our recent published review paper on oleogels for more information on the production methods of oleogels. Please see lines 42-43 in the revised version.
- The chemical composition of the substances used should be included.
Answer: In the section 2.1. Materials, available information about RBO, BW, refined CRW and EC has been given. EC had the following properties: 22 cP, 5% in toluene/ethanol 80:20 at 25oC, softening temperature 155°C and bought from Sigma-Aldrich. X-ray diffraction spectrums of all raw materials (BW, CRW and RBO and EC powder) all have been given separately in Figure 8.
- RBO stability at high temperature – has the authors checked whether the preparation procedure affects the unsaturated fatty acid residues included in the RBO triglyceride molecules? Indications for such effects are mentioned by the obtained DSC thermograms.
Answer: This is a good comment (and thank you), and we are trying to address and measure oil stability in a separate study. However, the objective of this current research is studying the physical properties of the prepared oleogels, but not the chemical stability.
- DSC results from the samples stored at 4 and 25 degs. What is the reason for the observed differences with respect to the melting enthalpies for the same sample stored at 4 and 25 deg? For example 4EC2CRW has enthalpy of 2.79 ± 0.38 J/g when stored at 25 deg and 2.15 ± 0.11 J/g when stored at 4 deg. As the sample are claimed to have the exact same composition – one would expect the same enthalpy or possible reversed behavior (if further crystallization takes place at lower T). This question also applies to many of the other results. Also, what is the reason for significantly higher crystallization enthalpies?
Answer: The 4 and 25 °C refer to two different storage temperatures, each of these oleogels have been stored overnight at either 4 or 25 °C, therefore significant differences were observed that are linked to their storage conditions. Storage at fridge (4 °C) and room temperature (25 °C) both are important in terms of oleogel preparation and application (after preparing oloegel mixtures, they are stored to set and solidify). Higher enthalpies are linked to higher wax crystal levels as a result of a thermal history with a storage time/temperature combination that better suits crystallization
- XRD part – crystals of EC are mentioned – the DSC does not support this claim, should be revised. “Enhanced crystallinity” claim should be revised.
Answer: please see Figure 7a, the sample marked as 8EC at the bottom. The peak at the 170-210 C range shows a melting peak on heating.
- Second p.2 in the XRD section – the sentence citing Yang et al. 2017 should be revised.
Answer: edited. Please see lines 632-635 of revised version.
- XRD – the spectra of pure RBO should be also included.
Answer: we have presented the X-ray spectra of all oleogels with different levels of wax. We have tried to get the X-ray spectrum of RBO too, but due to technical limitations with our X-ray equipment unfortunately we have not been able to record its spectrum and hence were unable to present.
Another minor comments:
- Line 29 “plastic fats” – what do you mean by this?
Answer: this has been changed from “plastic fats” to “solid fats” to make it more clearly.
- Lines 38-39 – references should be added
Answer: A reference has been added. (Martins et al., 2020)
- Lines 247-248 – how the stated cooling rates have been measured? Is the cooling rate linear throughout the whole time from the initial high T down to the low desired T?
Answer: The measurement method was stated in the method section (line 144-145). It was not a linear cooling rate and information given.
- Lines 240-253 – should be revised to state clearly what are the credible findings from these experiments. Pictures shown in Fig. 2a can be zoomed in to show more clearly the discussed structures.
Answer: This section has been edited and additional information on Figure2a images has been included. Please see the section 3. PLM part. Also, this was added “PLM images also show that in the absence of EC, the BW oleogel had thin needle like crystals, however the CRW oleogel had much thicker crystals. Moreover, the length of BW and CRW crystals were reduced in the presence of EC. “
- Lines 466-467 – the trends shown in Fig. 6b and 6d are not really “similar”.
Answer: it was meant that the gumminess index showed similar trends to hardness value (instead of cohesiveness). This has been revised (line 505-506)
- Lines 483-484 and 516-517 – these sentences should be revised.
Answer: these have been revised as per suggestion, to improve readability.
- Discussion about DSC results on the second p.1 – should be revised.
Answer: This has been revised, please see line 579-581.
- First sentence on the second p. 4 (above the conclusion section) – should be revised.
Answer: edited.
- The font of the “G” symbols within the manuscript should be unified.
Answer: edited
- Figure 3b – there are two curves with full pink circles.
Answer: edited
- Line 500 “more efficient that” -> than
Answer: the typo corrected
26.Second p.1 (DSC part) “Statically” -> statistically
Answer: Corrected

Reviewer 2 Report
Dear authors,
in a very good and accessible way, you described both the problem and the basics of the methodology and the results. However, there are several omissions in the work that need to be corrected.
line 123 - aberative PLM is without explanation - Polarized light microscope - it must be included here
line 123- PLM - please, also include here model, producer of the PLM
line 126 - Image J should heve reference: Abràmoff, M.D.; Hospitals, I.; Magalhaes, P.J.; Abràmoff, M. Image Processing with ImageJ.
line 180 - abberative DSC - Differential scanning calorimetry is without explanation
line 266 - aberrative CLSM is without explanation - confocal microscopy - please include the CLSM in your methodology, describe there model and produces of your CLSM
Figure 3 - I would enter in the drawings a/b/c/d/ the measurement temperatures - it would make the analysis easier for the reader
Author Response
Reviewer 2:
- line 123 - aberrative PLM is without explanation - Polarized light microscope - it must be included here
Answer: Added
- line 123- PLM - please, also include here model, producer of the PLM
Answer: The model, producer of the PLM was stated in line 145 (ECLIPSE Ci POL – Nikon, Japan)
- line 126 - Image J should heve reference: Abràmoff, M.D.;Hospitals, I.; Magalhaes, P.J.;Abràmoff, M. Image Processing with ImageJ.
Answer: A reference has been added in line 159. (Schneider et al., 2012)
- line 180 - abberative DSC - Differential scanning calorimetry is without explanation
Answer: The full name of DSC has now been stated in line 206 (i.e., 2.3.5 subtitle)
- line 266 - aberrative CLSM is without explanation - confocal microscopy - please include the CLSM in your methodology, describe there model and produces of your CLSM
Answer: Confocal laser scanning microscopy (CLSM) added; and the model and producers of CLSM was stated in line 160-161.
- Figure 3 - I would enter in the drawings a/b/c/d/ the measurement temperatures - it would make the analysis easier for the reader
Answer: Added
Reviewer 3 Report
The manuscript has an interesting objective and is in keeping with the journal to which it has been submitted. The introduction is very well written and the references are current and correctly selected. The methodology is well described, although the nomenclature of the samples is certainly confusing and in many parts of the text it is necessary to go back and check which sample is being discussed. Nevertheless, I understand that it must be difficult to choose a better one. In the results and discussion section, there may be some aspects that need to be clarified.
- Line 288. Perhaps indicating solid-like behaviour is too strict. One could say that it has more solid than liquid character.
- How have the critical strain values been determined?
- Line 357. "Oscillatory"
- Why has a characterization of the flow behavior not been made? Taking into account the analysis made in section 3.1 and the possible presence of yield stress, it would have been very interesting.
Author Response
Reviewer 3:
- Line 288. Perhaps indicating solid-like behaviour is too strict. One could say that it has more solid than liquid character.
Answer: using the term solid-like has been used according to the well accepted “solid-like” term used for samples with G’>G” as explained. On the other side when G”>G”, rheologists use the term “liquid like”. Please see below for more information on this terminology:
chrome-extension://efaidnbmnnnibpcajpcglclefindmkaj/https://www.tainstruments.com/pdf/literature/AAN016_V1_U_StructFluids.pdf
- How have the critical strain values been determined?
Added. According to the TA instrument application notes.
- Line 357. "Oscillatory"
Answer: Corrected
- Why has a characterization of the flow behavior not been made? Taking into account the analysis made in section 3.1 and the possible presence of yield stress, it would have been very interesting.
Answer: thank you, the oleogels studied in this research are semisolid. Therefore, their rheological behaviour properties were studied using oscillation (dynamic rheology), as rotational viscometry was not possible to perform, due to the rheological nature of the samples.